# Psychometric Evaluation of Women’s Knowledge of Healthcare Rights and Perception of Resource Scarcity during Maternity

**DOI:** 10.3390/healthcare12202045

**Published:** 2024-10-15

**Authors:** Claudia Susana Silva-Fernández, María de la Calle, María A. Suta, Silvia M. Arribas, Eva Garrosa, David Ramiro-Cortijo

**Affiliations:** 1Department of Biological & Health Psychology, Faculty of Psychology, Universidad Autónoma de Madrid, 28049 Madrid, Spain; 2Obstetric and Gynecology Service, Hospital Universitario La Paz, Universidad Autónoma de Madrid, 28049 Madrid, Spain; 3Centro de Investigaciones de la Fundación Oftalmológica de Santander (FOSCAL), Bucaramanga 680006, Colombia; 4Department of Physiology, Faculty of Medicine, Universidad Autónoma de Madrid, 28049 Madrid, Spain; 5Instituto Universitario de Estudios de la Mujer, Universidad Autónoma de Madrid, 28049 Madrid, Spain; 6Grupo de Investigación en Alimentación, Estrés Oxidativo y Salud Cardiovascular (FOSCH), Instituto de Investigación Sanitaria, Hospital Universitario La Paz, 28046 Madrid, Spain

**Keywords:** psychometric analysis, psychosocial factors, person-centered maternity, maternal rights, resources evaluation

## Abstract

**Background/Objectives**: Resources to cope with maternity and women’s participation are essential modulators of maternal well-being. Therefore, it is relevant that the psychosocial factors of woman be monitored during maternity to promote adequate healthcare. This study involved the design and the validation of two new tools that identify women’s knowledge of healthcare rights (MatCODE) and perception of resource scarcity (MatER) during pregnancy, labor and early postpartum; **Methods**: The content validity was carried out using the Aiken’s V coefficient and the content validity index (CVI-i) based on five experts. In addition, for the face validity, the pilot cohort was considered the INFLESZ scale. Finally, the questionnaires were applied to 185 women, which allowed to assess the construct validation by factorial and Rasch analysis. The divergent validity was also studied with validated psychological questionnaires; **Results**: MatCODE and MatER questionnaires received CVI-i and Aiken’s V > 0.80 values, and the INFLESZ demonstrated acceptable semantic understanding. The analysis confirms the unidimensionality of the questionnaires, with fit values for MatCODE of RMSEA = 0.113 [0.105; 0.122] and for MatER of RMSEA = 0.067 [0.063; 0.072]. The divergent validity showed significant and consistent correlations with the constructs assessed. For MatCODE, ω = 0.95 and α = 0.94, and for MatER, ω = 0.79 and α = 0.78; **Conclusions**: MatCODE and MatER are useful new tools for monitoring maternal healthcare, with adequate psychometric characteristics in the Spanish context.

## 1. Introduction

The person-centered maternity care allows a positive experience during pregnancy, childbirth, and postpartum, claiming women’s participation, evaluation, and intervention of their needs [1,2]. During obstetrical healthcare, the coping of the women can collapse by physical and psychosocial changes of motherhood [3]. Maternity healthcare management is essential to improve outcomes, since it facilitates adherence to clinical recommendations [4,5]. For women, the lack of knowledge about their sexual and reproductive rights increases the risk of acquiring a passive role during maternity and experiencing situations that affect their integrity and a loss of autonomy [6]. For the adaptation and well-being to motherhood, it is essential that woman perceive social–emotional support, procedural resources, physical care, and information [3,7]. In addition, clarity and confidence in solving the demands of motherhood are important considerations, since they have been related to greater adaptation to maternity [8]. The coping and empowerment of the woman during the healthcare process are also influenced by their resilience [3,9], affectivity [10,11], and beliefs [12].

The knowledge of rights during maternity care, beliefs about motherhood, and the perception of available resources can influence adjustment to maternity and women’s well-being. The knowledge of maternity rights (legal maternity leave, access to healthcare, and workplace accommodations) can help reduce stress and provide security during pregnancy and after childbirth. When women are aware of their rights, they may feel more empowered and supported by societal structures [13]. Informed women are likely to have healthier beliefs about motherhood. Maternity beliefs encompass attitudes and values a pregnant person holds regarding motherhood and its social role. Perception of adequate resources can also promote the belief that motherhood is manageable, and that help is available when needed, leading to less guilt or stress over seeking assistance. Social support (including at work and by health providers), financial stability, and knowledge about newborn and self-care in motherhood are key resources in a woman’s adjustment to motherhood [7,14]. Resilience refers to the ability to cope with and recover from challenges, including those associated with maternity [15]. During motherhood, women need to be resilient to balance work, breastfeeding, parenting, self-care, and other social roles [3]. When women know they have a safety net, they may feel more able to endure the challenges of motherhood with a sense of control.

The healthcare providers would apply evaluation tools to assess women’s knowledge of their rights and social, economic, emotional, and motivational resources. The recommendations in clinical evaluation consider guidelines for resolutions of the psychosocial issues and early prevention of health problems [9]. Therefore, the healthcare providers can apply tools to assess coping strategies [16], social support [17], work-home interactions [18], parenting stress [19], among others. However, a simple screening application to identify the psychosocial needs would be suitable. In the obstetric context, the Prenatal Biomedical Risk Scale [20] considers the psychosocial resources, but does not assess other variables such as motivation, work conflicts, and economic problems.

Considering the scarcity of tools that assess women’s resources and knowledge of their healthcare rights during the maternity period (pregnancy, labor, and postpartum), this study aims to develop and validate new tools related to the women’s knowledge of healthcare rights (MatCODE) and their perception of resource scarcity (MatER) during pregnancy, labor, and early postpartum in the Spanish context.

## 2. Materials and Methods

### 2.1. Ethical Aspect

This study was approved by the Research Ethics Committees of Universidad Autónoma de Madrid (Madrid, Spain; Ref.: CEI-112-2199, 22 January 2021). All women willing to participate were given an online information sheet, describing the aims of the study, and the informed consent form was signed in each case. Data collection was anonymous, and the database was blinded. In addition, this study adheres to the guidelines of the Standards for the Reporting of Diagnostic Accuracy Studies (STARD) [21] for assessment scale protocols.

### 2.2. Questionnaire Development: Items Generation and Scale Construction

The new questionnaires were generated to report relevant statements focusing on women’s knowledge of healthcare rights (MatCODE) and their perception of resource scarcity (MatER) during pregnancy, labor, and early postpartum. The perception of resource scarcity during maternity can impact a woman’s psychological and emotional well-being. When resources are perceived as limited, women may experience heightened stress and anxiety. This can influence their coping strategies, potentially leading to maladaptive responses. The MatER tool explores not only internal (psycho-emotional and cognitive processes) but also external resources (financial support, time, or social networks).

In both questionnaires, items were generated from a review of the literature [9] and discussion with maternity experts to enhance content validity. Drawing from previous clinical experiences in providing maternity healthcare, the researchers identified 11 items for MatCODE and 9 items for MatER. Items for the questionnaires were scored on a 5-point Likert scale, ranging in MatCODE from 1 = Strongly disagree to 5 = Strongly agree, and in MatER from 0 = Never to 4 = Always.

### 2.3. Expert Panel Review for Face and Content Validation

The procedure involved content validation by expert judgement and evaluation of face validity in a pilot cohort on a target population to assess the understandability of the questionnaires.

Content validity was assessed by a panel of five experts with more than ten years of experience in maternity research (two in clinical psychology, one in midwifery, one in medical physiology and one obstetrician), selected to provide different point-of-views on methodological issues. These experts were not part of the item generation and scale construction. The experts independently assessed the readability, understanding, and clarity of MatCODE and MatER. They also assessed the format of the questionnaires, determining whether an item evaluated what it was intended to evaluate and its importance within the construct. Each judge evaluated the content validity of each item and the instructions using a scale from 1 to 5 (where 1 was the lowest and 5 was the highest) based on three criteria. Therefore, the total possible score ranged from 3 to 15. The criteria were: (1) Language clarity—whether the item is clear and understandable in terms of its semantic and syntactic structure; (2) Relevance—whether the item is important for measuring the objective of the instrument; and (3) Coherence—whether the item aligns with the specific dimension of the research. Based on the experts’ scores, a content validity index (CVI-i) was calculated for each item, along with Aiken’s V coefficient [22]. This coefficient quantifies the relevance of each item regarding language clarity, relevance, and coherence domains based on the ratings of experts. The coefficient can have values between 0 and 1. The closer the value to 1, the greater validity [23]. Thus, value 1 is the highest possible value and indicates perfect agreement among the experts. To be considered as adequate, the value of the coefficient should be >0.8 [24]. In addition, the universal-CVI (UA-CVI) was calculated (the proportion of items on a scale that achieved a rating of 14 or 15 from all the experts) [13].

The face validity was performed in a target cohort to assess the understandability and acceptability of the items. This pilot test was carried out in 27 women, selected by non-probabilistic sampling based on the discretion of the research team. Participants had to meet the following inclusion criteria: age >18 years; having experienced a labor or a C-section within the last three years; having received healthcare for their last pregnancy, labor, and early postpartum; having been assisted in Spain; and having good comprehension of Spanish. The women were asked to assess the understandability of the questionnaires and suggest changes, if they deemed it appropriate, and therefore, the women could contact the research team to improve their understanding of the items. Additionally, the final version of the scale was evaluated using INFLESZ, a validated Spanish tool for evaluating text readability and ease of reading for healthcare services users (https://legible.es/blog/escala-inflesz/; accessed on 1 April 2021). It is based on Szigriszt Pazos’ perspicuity value as: 0–40, very difficult; 40–55, moderately difficult; 55–65, average difficulty; 65–80, easy; and 80–100, very easy [25].

### 2.4. Construct Validation Using Factorial Analysis

A cross-sectional study was designed to assess construct validity, reliability, divergent validity against the Resilience Scale (RS-14), the Positive and Negative Affect Schedule (PANAS), and the Maternity Beliefs Scale (MBS), and known-groups validation.

Women participants were selected by a non-probability sampling procedure. According to the theory of factor analysis, there must be at least 15 observations per item in the analyzed tool [26,27]. Selection of the sample was carried out through the research group’s social networks. This technique has demonstrated adequate recruitment response in other studies [28]. During the recruitment, 278 women were contacted. Inclusion and exclusion criteria were applied to the women contacted. The inclusion criteria were as follows: age > 18 years, having experienced vaginal labor or a C-section within the last three years, having received healthcare for the last pregnancy, labor, and early postpartum (up to 40 days after labor), and having good Spanish language understanding. The exclusion criteria were the inability to read/write in Spanish and home birth. Of the 278 women contacted, 185 met the inclusion and exclusion criteria. In the following data analysis, withdrawal criteria were considered (incorrect questionnaire, <75% of missing data, and the participant’s desire to leave the study). Finally, the sample consisted of 162 for MatCODE (withdrawal = 12.4%), and 143 for MatER (withdrawal = 22.7%). Adapted to STARD guidelines, a diagram depicting the flow of participants through the study is reported in Figure 1.

Data were collected from 1 September 2021 to 30 November 2023. A self-administered online tool was prepared by Qualtrics (https://www.qualtrics.com/es/; accessed on 15 July 2021). First, it obtained sociodemographic and obstetric variables, as well as responses to psychometric tests used for divergent validity. Second, responses to the MatCODE and MatER questionnaires were collected.

The variables collected in the first part were age (years), education level, civil status (single/unmarried vs. any type of relationship), employment status (active working vs. unemployed), number of deliveries, type of last labor (vaginal vs. C-section), whether the last pregnancy was planned or desired (yes/no), use of assisted reproduction techniques (yes/no), multiple pregnancy (yes/no), gestational age in the last pregnancy (weeks), premature labor (<37 weeks of gestation), last labor by lithotomy (yes/no), presentation of a birth plan (yes/no), and adverse outcomes (yes/no) during pregnancy, labor, or early postpartum.

The second part included: (1) the MatCODE questionnaire, a new tool designed to assess the level of knowledge women have of their healthcare rights during pregnancy, labor, or postpartum. MatCODE is an 11-item scale scored on a Likert-type format from 1 = strongly disagree to 5 = strongly agree. Higher scores on MatCODE indicate a greater awareness of their healthcare rights. (2) the MatER questionnaire, a new tool designed to assess the woman’s perception of her pregnancy, labor, or early postpartum resources. MatER is a 9-item scale scored on a Likert-type format from 0 = never to 4 = always. Higher scores on MatER indicate a lower perception of resources of the woman. The Spanish version of questionnaires can be found in the Appendix B (MatCODE) and Appendix C (MatER).

To assess divergent validity, women responded to (1) the Resilience Scale (RS-14) [29]. This scale assesses the ability to cope with daily difficulties. It was used in the 14-items version. The higher the score, the greater the woman’s ability to cope with the problems of everyday life. Other studies reported a Cronbach’s α coefficient of 0.88. (2) The Positive and Negative Affect Schedule (PANAS) [30]. One of the most widely used scales to measure emotion. This scale has 20 items, with 10 items measuring positive affect (e.g., excited, inspired) and 10 items measuring negative affect (e.g., upset, afraid). For the positive score (PANAS+), a higher score indicates more of a positive affect. For the negative score (PANAS−), a lower score indicates less negative affect. PANAS obtained a Cronbach’s α coefficient scores ranging from 0.86 to 0.90 for the positive dimension (PANAS+) and 0.84 to 0.87 for the negative dimension (PANAS−) [31]. (3) The Maternity Beliefs Scale (MBS). This scale identifies specific beliefs that women hold related to maternity, based on the Rational Emotive Behavior Theory. MBS has 13 items, clustered into two subscales: maternity as a sense of life (MBS-life) and maternity as a social duty (MBS-social). The higher the score, the higher the woman’s belief in the indicated domain. The global Cronbach’s α coefficient was 0.93, with MBS-life = 0.92 and MBS-social = 0.83 [32].

### 2.5. Data Analysis

A descriptive analysis of the variables and items was conducted. Categorical variables were expressed as relative frequencies (%); quantitative variables were expressed as mean ± standard error of mean (SEM). Symmetry and kurtosis were calculated for each item. In addition, the Relative Difficulty Indexes (RDI) and the normed Measure of Sampling Adequacy (MSA) values were calculated [33]. RDI evaluates the position of the items; nearly 75% of item values should fall between 0.40 and 0.60. Lower MSA values indicate that item randomly behaves, with 0.50 as the cut-off limit (inappropriate item with non-discrimination) [33].

**Construct validity by factor analysis**. The suitability of data for a factor analysis was assessed with the Kaiser–Meyer–Olkin index (KMO) and Bartlett’s statistic. KMO ≥ 0.75 was considered adequate, and *p* ≤ 0.05 was considered statistically significant for Bartlett’s statistic. A confirmatory factor analysis (CFA) was carried out based on the obtained dimensional model.

Following the González-de la Torre et al. approach [13], the suitability of the factorial solution was assessed by the Root Mean Square of Residuals (RMSR; values < 0.08 are generally considered a good fit) [34] and associated Kelley’s criterion [35]. The Root Mean Square Error of Approximation (RMSEA; values < 0.05 were considered a good fit, and values between 0.05 and 0.08 were considered reasonable fits) [34], the estimated Non-Centrality Parameter (NCP), and associated *p*-value (*p*), testing whether the value corresponds to a non-central distribution. The Non-Normed Fit Index (NNFI), the Comparative Fit Index (CFI), and the Adjusted Goodness-of-Fit Index (AGFI) were also evaluated. NNFI and CFI values ≥ 0.95 and AGFI values > 0.90 were considered a good fit of the model [26]. The Common part Accounted for (CAF) by a common factor model expresses the extent to which the common variance in the data is captured by the model [36]. If the CAF value is close to 0, it means that more factors should be extracted. The Bayesian Information Criterion (BIC; degree of parsimony index) favors lower values as indicators of better fit. The Weighted Root Mean Square Residual (WRMR), with values < 1.0, has been recommended to represent good fit [37].

The matrix rotation of the items was applied in all solutions by a Promax oblique rotation. The number of factors to be retained was established through a parallel analysis, and the communalities of the item were calculated. The 95% confidence intervals [95%CI] were calculated for the item scores and the model measures. To evaluate the dimensionality, the Unidimensional Congruence (UniCo), Explained Common Variance (ECV), and Mean of Item Residual Absolute Loadings (MIREAL) indices were used [38]. UniCo > 0.95, ECV > 0.85, and MIREAL < 0.30 indicate that the data can be considered as essentially unidimensional.

**Construct validity by Rasch model**. The MatCODE score was adapted from a 1–5 range to a 0–4 range, while MatER already showed this codification. Item fit was estimated by outfit Unweighted Mean Square fit statistic (UMS) and infit Weighted Mean Square Fit statistic (WMS). Fit indices between 0.8 and 1.2 were considered as a good fit, while values between 0.5 and 1.5 were considered acceptable [39]. The quality was established by reliability (measure to order item difficulty), with desirable values > 0.7. In addition, to establish the reliability of MatCODE and MatER, the Alpha (α) and Omega (ω) coefficients were calculated. The divergent analysis explored correlations between MatCODE and MatER with related psychometric variables. First, the global MatCODE and MatER scores were calculated by summing the Liker scores. Second, the validated psychometric scales and MatCODE and MatER global score were standardized, and the Spearman’s correlation coefficient (ρ) was used, with statistical significance considered at *p* < 0.05.

**Known-groups validation**. To explore the association between the different variables and the MatCODE and MatER global scores, an inferential analysis was conducted, comparing groups of women likely to have experienced obstetric vulnerability according to several aspects described in the literature [9]. The non-parametric Mann–Whitney U test was used to compare groups. A *p* ≤ 0.05 was considered statistically significant. The effect size was calculated using the Hedges’ g.

#### Statistical Software

The descriptive and inferential analyses were performed using R software within the RStudio interface (version 2022.07.1+554, 2022, R Core Team, Vienna, Austria) using *rio*, *dplyr*, *compareGroups*, *devtools*, *psych* [40], and *lavaan* [41] packages. For reliability analysis, the *eRm* [42] and *TAM* [43] packages were used. In addition, the factor analysis and index evaluation were carried out with the free software FACTOR© Release version 12.04.05 ×64 bits (https://psico.fcep.urv.cat/utilitats/factor/Download.html; accessed on 1 September 2021).

## 3. Results

### 3.1. Content Validity

All items in both questionnaires (MatCODE and MatER) received CVI-i > 0.80 values. Appendix A shows the scores assigned to each item by the experts, as well as the CVI-i values. In MatCODE, the UA-CVI was 54.5%, and in MatER, it was 77.8%.

The pilot cohort did not report difficulties in understanding any of the items. Therefore, item modifications were not introduced. According to INFLESZ, the perspicuity score for MatCODE was 56.59, and for MatER, it was 57.85, both indicating average readability difficulty, which corresponds to a text with a normal level of readability.

### 3.2. Descriptive Analysis of Sample and Items

The women’s ages were 28.5 ± 0.47 years (range: 18–42 years), with 89 (54.6%) being primiparous and 74 (45.4%) multiparous. Regarding education levels, 9 (5.5%) had primary education, 77 (47.2%) had secondary education, and 77 (47.2%) completed university studies. In total, 29 (17.8%) of the women were unmarried/single, and 134 (82.2%) were married or in some form of sentimental relationship. Regarding employment situation, 90 (55.2%) of the women were actively working and 73 (44.8%) were unemployed.

The last pregnancy was planned for 86 (52.8%) of the women, and it was desired by 125 (76.7%). The gestational age was 38.7 ± 0.12 weeks (range: 29–41.5 weeks), with 29 (17.8%) cases of premature labor. C-section were performed in 79 (48.5%) women, and assisted reproduction techniques were used in 7 (4.3%) case, 3 cases of multiple pregnancy 3 (1.8%). The lithotomy delivery position was performed in 139 (85.3%) cases.

Pregnancy complications were presented in 52 (31.9%) women, complications during labor by 28 (17.2%), and during early postpartum period by 26 (16.0%). Finally, women were asked if they had presented a birth plan and whether the plan had been followed. Many of the women (160; 98.8%) did not present a birth plan.

Regarding the MatCODE and MatER questionnaire responses, Table 1 shows a descriptive analysis, the symmetry and kurtosis of the items. In MatCODE, the RDI was >0.60 indicating a normal-range test and an optimal pool of items. In addition, the normed MSA values were >0.50, suggesting that the items measured the same domain as the item pool. In MatER, RDI ranged between 0.20 to 0.40, indicating that several items were placed within the extreme quartiles. The normed MSA was >0.50 and items measured the same domain.

### 3.3. Construct Validity by Factor Analysis

#### 3.3.1. Confirmatory Factor Analysis

A confirmatory factor analysis was performed based on the proposed 11-item MatCODE questionnaire and 9-items MatER questionnaire (correlation matrices are shown in Appendix A). In MatCODE, the KMO and Bartlett’s statistic indicated an acceptable fit for the sample (KMO = 0.901 [0.827; 0.915]; *p* < 0.001 Bartlett’s test). The one-factor solution showed 65.1% explain variance, indicating a single-factor result from the parallel analysis. In MatER, the KMO and Bartlett’s test showed a good fit (KMO = 0.842 [0.718; 0.850]; *p* < 0.001 for Bartlett’s test). Additionally, the one-factor solution showed 37.9% explained variance. The robust goodness of fit for MatCODE and MatER is reported in Table 2.

#### 3.3.2. Exploratory Factor Analysis

For MatCODE, the unidimensionality assessment showed UniCo = 0.987 [0.973; 0.997], ECV = 0.896 [0.848; 0.936], and MIREAL = 0.242 [0.174; 0.299]. For MatER, the values were UniCo = 0.945 [0.913; 0.981], ECV = 0.793 [0.728; 0.850], and MIREAL = 0.187 [0.176; 0.184]. These results support the one-dimensionality of the scales. The results of the factor loadings for each item and their commonality can be seen in Table 3.

#### 3.3.3. Reliability of the Instruments

In MatCODE, the reliability of items was 0.832, and in the MatER was 0.711, which indicated acceptable reliability. Overall reliability was explored by the omega (ω) and Cronbach’s alpha (α) coefficients. In MatCODE was ω = 0.95 and α = 0.94 [0.93; 0.96] and in MatER was ω = 0.79 and α = 0.78 [0.73; 0.83]. The infit WMS and outfit UMS values are shown in Table 4. Infit WMS values indicated good or acceptable fit for all items, except item 9 for MatCODE and item 5 for MatER. Outfit UMS values showed an acceptable fit for all items in MatCODE. Item 5 for MatER also obtained a higher UMS score. The removal of item 5 from MatER did not modify the divergent analysis (Appendix A) or known-groups validation (Appendix A).

### 3.4. Divergent Validity

The total score in the resilience scale, PANAS+ and PANAS−, and MBS-life and MBS-social is reported in Table 4. MatCODE and MatER show a significantly negative correlation (ρ = −0.20 [−0.35; −0.03]; *p* = 0.019). In addition, the MatCODE and MatER scores show positive and negative correlations with resilience score, respectively. Similarly, significant correlations were shown with PANAS+. In addition, MatER, but not MatCODE, shows positive correlations with PANAS−. Although the MBS-life score did not show any statistical correlation with MatCODE, MatCODE, but not MatER, showed a statistically negative correlation with MBS-social dimension (Table 5).

### 3.5. The Known-Groups Validation

The total MatCODE (range: 11–55) and MatER (range: 0–36) scores are calculated by adding individual item scores. The mean score recorded in the sample for the MatCODE was 47.10 ± 0.67 (Min = 11; Max = 55), and for the MatER was 10.60 ± 0.55 (Min = 0; Max = 30).

The MatER score, but not MatCODE, was significantly negatively correlated with age (ρ = −0.169; *p* = 0.044). The MatCODE and MatER scores were not significantly different between pregnancy and last labor. However, MatCODE and MatER scores were statistically different between women who desired the pregnancy, and MatER score was also different between women who planned pregnancy and developed postpartum complications (Table 6).

## 4. Discussion

Given the importance of supporting women in motherhood, it is crucial to assess healthcare rights and perceptions of resources during maternity. In conclusion, both the MatCODE and MatER tools were contextually validated. Both questionnaires passed through the psychometric steps to guarantee their reliability. According to Pedrosa et al. [44], the content validity was carried out by interdisciplinary experts with several years of experience, representatives of both sexes, who did not participate in the design of the questionnaires, allowing a comprehensive and objective analysis of the tools. In addition, the content validity showed consistency and representativeness of the items aligned with the purpose of the tools’ assessment. CVI-i was used to maintain the results with Aiken’s V coefficient, as it is suitable for processes involving fewer than six experts [44]. Thus, the estimation of agreement was verified without the effect of the number of experts. Furthermore, face validity reinforced the acceptance of the items with a slight improvement in the semantics of the items, which maintains the validity of the tools [45].

The divergent validity of MatCODE showed consistency with previous findings on knowledge, empowerment, and self-esteem [8,46]. The positive affective effects and greater adaptative capacity in the face of adversity were expected. Furthermore, the concept of maternity as a social duty, where the identity of motherhood is shaped by a paternalist perspective and woman are seen as passive subjects [12], can be consistent with MatCODE scores. A woman who perceives motherhood in this way may be less aware of her healthcare rights and experiences decreased empowerment. Similar to MatER, where a higher score means fewer resources, its negative correlations suggest that fewer resources are associated with a greater risk of affective vulnerability [47]. The assessment variables of both questionnaires demonstrated an association with components that identify psychosocial vulnerability during maternity.

The factor analysis confirms the unifactorial design of both questionnaires. This analysis revealed a greater explanatory variance for MatCODE, demonstrating the robustness of its items in representing the construct. The results related to MatER may be attributed to a lower representativeness of resource differences (presence or absence) within the women, potentially due to the reduced sample size. The items functioned as dependent factors that explain the latent variable of the questionnaires. Both questionnaires had an adequate model fit. However, the item 5 of MatER showed low saturation of the construct. Similarly, the Rasch analysis demonstrated tendency toward randomness. According to Lloret-Segura et al., these items should be modified or eliminated for the final proposal of the questionnaire [27]. However, our results did not show any change in the validation analyses after removing them. This may be because the whole model showed good robustness indices in the exploratory and confirmatory analysis. In addition, consistent with findings from other authors, the known-groups validation showed that younger age [48], unplanned pregnancy [49], and obstetrical complications [50] were associated with an increased perception of difficulties in coping to maternity, specifically in psychosocial resources.

When an unplanned pregnancy occurs, pregnant people may have greater difficulty to accept motherhood. According to Martínez-Galiano & Delgado-Rodríguez [46], a woman’s participation in aspects related to her gestational health depends directly on her awareness of the event. Therefore, it is likely that a woman has less awareness in the case of an unplanned pregnancy, leading to lower acceptance, a more negative attitude, and reduced participation in her healthcare.

In summary, both new questionnaires, MatCODE and MatER, are valid and reliable assessment tools within Spanish context and can be useful for complement screening and psychosocial monitoring during pregnancy, childbirth, and postpartum. These tools would impact the improvement of women’s well-being and the person-centered maternity healthcare, since they contribute to individual profiling of psychosocial needs and guide care to prevent health issues. Specifically, MatER helps women recognize the limited resources available to cope with the changes of motherhood, while MatCODE highlights the need for education about their rights and empowerment throughout the healthcare process. The questionnaires are complementary can integrate analyses with other components that also affect well-being, such as obstetric conditions [51], mental history, or lifestyles [47], among others. Furthermore, if the scores of the questionnaires indicate any need for psychosocial intervention, it is recommended to extend the assessment with other tools, such as clinical interviews or tests for identified risk variables.

A limitation of the study is the small sample size and its restriction to the Spanish-context, suggesting the need for validation in other sociocultural contexts with larger samples. This aligns with Lloret-Segura et al., who highlight that the number of elements analyzed and their communality indicate the need for a larger sample [27]. Since MatCODE and MatER focus on psychosocial variables that may depend on sociocultural components, it is important to delimit these components in future psychometric research.

## 5. Conclusions

During maternity, coping and suitable support are importance to fit women for motherhood social roles. Healthcare providers need tools to assess health rights and women’s perception of resources to apply adequate intervention. MatCODE and MatER are tools with adequate psychometric properties, reliable and useful for measuring women’s knowledge about their healthcare rights and perception of resources during maternity in Spanish-speaking context. Additionally, the questionnaires are easy for women to complete and for health staff to extract scores. MatCODE and MatER can guide healthcare providers on psychosocial interventions for better fit outcomes during maternity. It would be useful to validate these tools in other cultural contexts and explore their relationship with obstetric violence.

## Figures and Tables

**Figure 1 healthcare-12-02045-f001:**
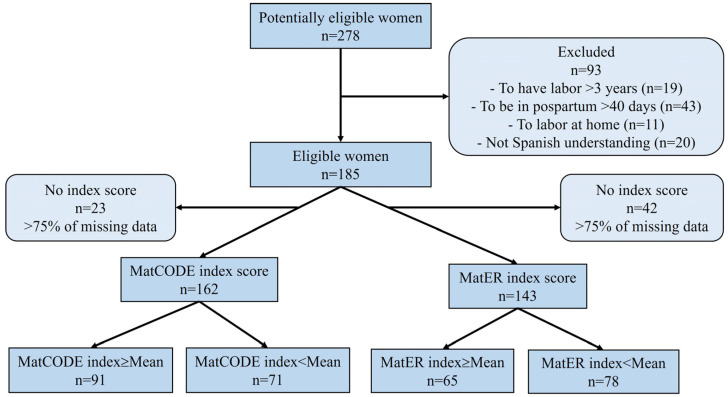
Flow for the study design adapted from STARD guidelines. MatCODE: Knowledge of healthcare rights during pregnancy, labor, and early postpartum tool; MatER: Perception of resource scarcity during pregnancy, labor, and early postpartum tool. The MatCODE mean was 47.1, and the MatER mean was 10.6.

**Table 1 healthcare-12-02045-t001:** Descriptive analysis of the items in the MatCODE and MatER questionnaires.

MatCODE	Mean	Variance	Skewness	Kurtosis	RDI	Normed MSA
Item 1	4.142	1.097	−1.520	1.990	0.785	0.865
Item 2	3.951	1.393	−1.151	0.480	0.738	0.893
Item 3	4.333	0.691	−1.534	2.804	0.833	0.935
Item 4	4.105	1.045	−1.259	1.159	0.776	0.933
Item 5	4.272	0.815	−1.577	2.830	0.818	0.925
Item 6	4.290	0.897	−1.575	2.357	0.823	0.890
Item 7	4.340	0.866	−1.789	3.384	0.835	0.956
Item 8	4.463	0.854	−2.230	4.946	0.866	0.886
Item 9	4.290	1.231	−1.740	2.229	0.823	0.895
Item 10	4.457	0.804	−2.119	4.608	0.864	0.885
Item 11	4.463	0.779	−2.080	4.584	0.866	0.854
**MatER**	**Mean**	**Variance**	**Skewness**	**Kurtosis**	**RDI**	**Normed MSA**
Item 1	1.364	1.406	0.430	−0.671	0.341	0.866
Item 2	0.755	1.150	1.153	0.254	0.189	0.846
Item 3	1.273	1.457	0.543	−0.568	0.318	0.831
Item 4	0.797	1.113	1.062	0.168	0.199	0.801
Item 5	0.916	2.021	1.296	0.144	0.229	0.773
Item 6	1.622	1.843	0.153	−1.198	0.406	0.886
Item 7	1.091	1.523	0.725	−0.615	0.273	0.851
Item 8	1.692	1.514	0.174	−0.841	0.423	0.828
Item 9	1.091	1.341	0.719	−0.488	0.273	0.871

RDI: Relative Difficulty Indexes; MSA: Measure of Sampling Adequacy. MSA < 0.50 indicates inappropriate item with non-discrimination.

**Table 2 healthcare-12-02045-t002:** Robustness of the model and distribution of residuals for the MatCODE and MatER questionnaires.

	MatCODE	MatER
RMSEA	0.113 [0.105; 0.122]	0.067 [0.063; 0.072]
NCP	17.710 (*p* = 0.930)	9.585 (*p* = 0.843)
NNFI	0.966 [0.956; 0.972]	0.949 [0.896; 0.982]
CFI	0.973 [0.965; 0.977]	0.962 [0.922; 0.987]
BIC	246.199 [238.344; 257.125]	133.484 [130.534; 137.375]
RMSR	0.080 [0.05; 0.10]	0.093 [0.08; 0.09]
Kelley’s criterion	0.079	0.084
WRMR	0.096 [0.05; 0.13]	0.094 [0.08; 0.09]
AGFI	0.987 [0.957; 0.994]	0.982 [0.940; 0.982]
CAF	0.432	0.498

MatCODE: Knowledge of healthcare rights during pregnancy, labor, and early postpartum tool; MatER: Perception of resource scarcity during pregnancy, labor, and early postpartum tool; RMSEA: Root Mean Square Error of Approximation; NCP: estimated Non-Centrality Parameter; NNFI: The Non-Normed Fit Index; CFI: the Comparative Fit Index; BIC: Bayesian Information Criterion; RMSR: Root Mean Square of Residuals; WRMR: Weighted Root Mean Square Residual; AGFI: Adjusted Goodness-of-Fit Index; CAF: Common part Accounted.

**Table 3 healthcare-12-02045-t003:** The dimensionality assessment, communality and factor loading of the one-dimensional model for MatCODE and MatER questionnaires.

MatCODE	I-UniCo	Communality	Rotated Factor
Item 1	0.988 [0.819; 1.000]	0.563	0.711
Item 2	0.978 [0.816; 1.000]	0.577	0.710
Item 3	0.999 [0.979; 1.000]	0.655	0.780
Item 4	0.991 [0.837; 1.000]	0.667	0.787
Item 5	1.000 [0.972; 1.000]	0.710	0.821
Item 6	1.000 [1.000; 1.000]	0.683	0.812
Item 7	1.000 [0.991; 1.000]	0.686	0.811
Item 8	0.944 [0.633; 0.993]	0.566	0.727
Item 9	0.969 [0.818; 0.998]	0.578	0.733
Item 10	0.998 [0.980; 1.000]	0.705	0.842
Item 11	0.995 [0.964; 1.000]	0.773	0.881
**MatER**	**I-UniCo**	**Communality**	**Rotated Factor**
Item 1	1.000 [1.000; 1.000]	0.367	0.531
Item 2	0.994 [0.644; 1.000]	0.387	0.540
Item 3	1.000 [1.000; 1.000]	0.480	0.640
Item 4	0.536 [0.187; 0.924]	0.309	0.470
Item 5	0.999 [0.919; 1.000]	0.159	0.310
Item 6	0.999 [0.888; 1.000]	0.229	0.406
Item 7	0.989 [0.578; 1.000]	0.429	0.593
Item 8	0.991 [0.696; 1.000]	0.525	0.696
Item 9	1.000 [0.986; 1.000]	0.523	0.675

I-UniCo: Unidimensional Congruence of the item. Communalities is the proportion of each item variance that can be explained by the unique factor. The Promax oblique rotation was applied.

**Table 4 healthcare-12-02045-t004:** Fit values of the items by the joint maximum likelihood estimation method in the rasch analysis.

MatCODE	Infit WMS	Outfit UMS	MatER	Infit WMS	Outfit UMS
Item 1	1.26	1.29	Item 1	0.88	0.87
Item 2	1.26	1.20	Item 2	0.97	0.91
Item 3	0.85	0.65	Item 3	0.85	0.86
Item 4	0.93	0.85	Item 4	0.99	0.95
Item 5	0.78	0.73	Item 5	1.74	1.78
Item 6	0.96	1.04	Item 6	1.19	1.22
Item 7	0.96	0.66	Item 7	0.98	0.99
Item 8	1.42	0.81	Item 8	0.76	0.77
Item 9	1.58	1.23	Item 9	0.80	0.79
Item 10	1.05	0.70			
Item 11	0.84	0.48			

UMS: Unweighted Mean Square fit index; WMS: Weighted Mean Square fit index. Fit index between 0.8–1.2 meant a good fit and between 0.5–1.5 meant an acceptable fit.

**Table 5 healthcare-12-02045-t005:** Correlation coefficient between MarCODE and MatER scores with validated psychometric scales.

	Mean ± SEM (n = 162)	MatCODE	MatER
RS-14	82.08 ± 1.03	0.17 [0.01; 0.33] *p* = 0.037	−0.32 [−0.47; −0.17] *p* < 0.001
PANAS+	36.67 ± 0.60	0.24 [0.08; 0.39] *p* = 0.004	−0.46 [−0.58; −0.33] *p* < 0.001
PANAS−	23.83 ± 0.68	−0.05 [−0.21; 0.12] *p* = 0.575	0.49 [0.35; 0.60] *p* < 0.001
MBS-life	17.27 ± 0.58	−0.07 [−0.23; 0.09] *p* = 0.388	0.03 [−0.13; 0.20] *p* = 0.664
MBS-social	8.00 ± 0.29	−0.20 [−0.35; −0.04] *p* = 0.018	0.06 [−0.10; 0.22] *p* = 0.469

Data show Pearson’s correlation and 95% confidence intervals [CI]. SEM: standard error of mean; RS-14: the Resilience Scale; PANAS+: the Positive and Negative Affect Schedule positive score; PANAS−: the Positive and Negative Affect Schedule negative score; MBS-life: the Maternity Beliefs Scale maternity as a sense of life domain; MBS-social: the Maternity Beliefs Scale as a social duty domain. A *p*-Value (*p*) < 0.05 was considered statistically significant.

**Table 6 healthcare-12-02045-t006:** Descriptive and bivariate analysis for known-groups validation.

		MatCODE			MatER		
		Mean ± SEM	*p*	Effect Size	Mean ± SEM	*p*	Effect Size
Parity	Primiparous	47.05 ± 0.96	0.859	0.02	11.19 ± 0.76	0.242	0.19
	Multiparous	47.18 ± 0.96			9.91 ± 0.81		
Civil status	Single/unmarried	45.83 ± 1.54	0.200	0.18	12.73 ± 1.72	0.070	0.40
	In relationship	47.38 ± 0.75			10.13 ± 0.55		
Work situation	Unemployed	46.56 ± 1.07	0.415	0.12	11.47 ± 0.87	0.075	0.24
	Active working	47.55 ± 0.87			9.86 ± 0.71		
Planned pregnancy	Yes	47.08 ± 1.00	0.972	0.01	9.08 ± 0.65	0.003	0.50
	No	47.12 ± 0.90			12.33 ± 0.88		
Desired pregnancy	Yes	47.47 ± 0.75	0.035	0.53	9.94 ± 0.54	0.002	0.77
	No	42.89 ± 2.44			8.73 ± 2.00		
Last labor	C-section	46.08 ± 1.04	0.097	0.23	11.25 ± 0.81	0.210	0.18
	Vaginal	48.06 ± 0.87			10.06 ± 0.76		
Pregnancy complications	Yes	46.54 ± 1.24	0.529	0.10	11.80 ± 1.06	0.151	0.26
	No	47.37 ± 0.81			10.05 ± 0.64		
Labor complications	Yes	46.74 ± 1.23	0.270	0.05	11.78 ± 1.34	0.234	0.22
	No	47.18 ± 0.77			10.33 ± 0.61		
Postpartum complications	Yes	46.88 ± 1.35	0.787	0.03	14.14 ± 1.38	0.005	0.65
	No	47.15 ± 0.74			9.96 ± 0.59		

Data show mean ± standard error of mean (SEM). Statistical significance was established as (*p*) ≤ 0.05 by Mann–Whitney U-test. An effect size (Hedges’s g) < 0.2 indicates a small effect, 0.5 a medium effect, and 0.8 a large effect.

## Data Availability

The original contributions presented in the study are included in the article/Appendix A, further inquiries can be directed to the corresponding author.

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
