# Peer review of "Psychometric Evaluation of Women’s Knowledge of Healthcare Rights and Perception of Resource Scarcity during Maternity"

_healthcare, 2024, doi:10.3390/healthcare12202045_

Round 1
Reviewer 1 Report
Comments and Suggestions for Authors
Thank you for the opportunity to review your article. It would be helpful to identify pregnant people who are unaware of their rights or do not know where to find resources. A short survey would help providers identify patients needing more assistance during pregnancy.
Editing the grammar of the article will greatly enhance it, making it easier to read and understand.
Try to use neutral pronouns or use pregnant person
Strengthen the arguments for the reason for developing the tools.
Line 79 – What is meant by databases?
Line 97-98: I wonder why the genders of the experts were identified
Lines 103-105: Please clarity how the scale of 1 to 5 and the criteria of min to max were used. I don’t understand how this was done.
Line 107: How do coherence and clarity differ?
Lines 109-115: Does the coefficient in lines 109 and 111 refer to Aiken’s V? How is the CV-I calculated and interpreted? It is stated that total agreement is needed for an item to be valid, but then it is indicated that > .80 is considered adequate. These two statements are in opposition to each other.
Lines 191-206: Please explain and justify using the PANAS , MBS, and RS-14 for divergent validity. How do they relate to the MatCODE and MatER?
Lines 212 to 215: Please provide more information about .40 and .60, and .50. What do these values represent?
Line 215: Please describe the data analyses for content and face validity.
Line 255: The analyses for known groups are described, but it is not included in the methods section. The divergent validity is described in the methods section, but not the data analysis section. Please describe both types of validity in both sections.
Line 264: Reliability is mentioned for the first time. Please describe in the methods and data analysis section.
Line 272: UA-CV was mentioned in the methods section but is not described in the data analysis section.
Line 275: Based on the information in Lines 127—129, the average difficulty reading score is a perspicuity value between 55-65. Is this equivalent to a reading level of 5th grade? I wonder if it would be better if the survey were in the easy range.
Section 3.31: Consider putting these values in a table and summarize in text. It is challenging to read all these results.
Line 336: Please consider inserting a heading for reliability
Line 337: Please provide information on what is an acceptive value for a reliability coefficient in the data analysis section.
Line 385: Shouldn’t the instrument be both reliable and valid?
Comments on the Quality of English Language
Please have an English expert revise your manuscript. It wasn't easy to understand.
Author Response
Thank you for the opportunity to review your article. It would be helpful to identify pregnant people who are unaware of their rights or do not know where to find resources. A short survey would help providers identify patients needing more assistance during pregnancy.
Response: Thank you for your kind words and your time spend reviewing our article. Your comments were carefully reviewed and were answer point-by-point below.
Editing the grammar of the article will greatly enhance it, making it easier to read and understand. Try to use neutral pronouns or use pregnant person
Response: Thank you for this recommendation. All the text was reviewed to improve the comprehensibility.
Strengthen the arguments for the reason for developing the tools.
Response: New inside was written in the introduction section to clarify the importance to develop new tools in this field (lines 76-80).
- Line 79 – What is meant by databases?
Response: In this type of manuscript, databases refer to organized collections of information that we used to find relevant results. It describes the sources used to support our study, including the age of the women, response of each question, etc. This database was used to perform the statistical analysis.
- Line 97-98: I wonder why the genders of the experts were identified.
Response: It was considered relevant to show impartiality and the gender equality due to this study involves a gender analysis phenomenon. However, we agree with this comment and the gender of the experts was deleted.
- Lines 103-105: Please clarity how the scale of 1 to 5 and the criteria of min to max were used. I don’t understand how this was done.
Response: The judges evaluated each item of the questionnaire in 3 dimensions (language/relevance/coherence). The judges could award 1 to 5 points in each dimension to each item. If a judge scored an item with a 1 in all three dimensions, this item would obtain a minimum score of 3, if the judge scored it in all three dimensions with a 5, the item would obtain a maximum score of 15. The text was re-written to clarify it (lines 118-123).
- Line 107: How do coherence and clarity differ?
Response: Clarity and coherence are related but refer to different aspect. Clarity refers to how easy it is to understand something, it means the language is simple, direct, and free from confusion. Coherence refers to how well different parts of something fit together logically. If a message is coherent, it means all its parts make sense. In a research context, a question is coherent if it connects logically with the overall topic or objective of the research, and it is clear if is successfully informs what is intended.
- Lines 109-115: Does the coefficient in lines 109 and 111 refer to Aiken’s V? How is the CV-I calculated and interpreted? It is stated that total agreement is needed for an item to be valid, but then it is indicated that > .80 is considered adequate. These two statements are in opposition to each other.
Response: Thank you for this great appreciation. Content Validity Index (CVI-i) and Aiken's V coefficient are 2 methods used to measure the content validity of an instrument. Both assess whether the items adequately cover the concept studied, but they differ in approach (DOI: 10.5944/ap.10.2.11820). The CVI-i assesses if the judges agree that an individual item is valid. To calculate is applied the mean of the experts in a particular item. Higher values (close to 1) indicate greater agreement among experts and stronger content validity.
Aiken’s V is used to quantify how much agreement exists among raters when they evaluate an item on a continuous scale. Aiken’s V considers both the number of judges, and the scale points used.
A value closer to 1 indicates a higher level of agreement among experts that an item is relevant. Aiken’s V is preferred when a more nuanced interpretation of agreement is needed, while CVI-i is often simpler to calculate and interpret. The reference [19] deeply discuss this information.
- Lines 191-206: Please explain and justify using the PANAS, MBS, and RS-14 for divergent validity. How do they relate to the MatCODE and MatER?
Response: Thank you for this value comments. We will implement a need paragraph related knowledge of maternity rights and perceived resources during maternity and their relationship with affectivity, maternity beliefs and resilience (lines 53-69).
- Lines 212 to 215: Please provide more information about .40 and .60, and .50. What do these values represent?
Response: These are the cut-off points described to consider that the difficulty and adequacy of the items to the construction is relevant. The mathematical modeling and development of the formulas can be found in more detail in new reference [35].
- Line 215: Please describe the data analyses for content and face validity.
Response: The data analyses were described in line 111 to 124 for content validity and 132 to 144 for face validity.
- Line 255: The analyses for known groups are described, but it is not included in the methods section. The divergent validity is described in the methods section, but not the data analysis section. Please describe both types of validity in both sections.
Response: The divergent validity was described in material section (lines 207-222) and interpreted in the result section 3.4. The known-groups validity was described in the methods section (lines 271-276) and interpret in the result section 3.5.
- Line 264: Reliability is mentioned for the first time. Please describe in the methods and data analysis section.
Response: The reliability was mentioned for the first time in the line 146. In addition, in the line 263-266 was explained how to be calculated and interpretated.
- Line 272: UA-CV was mentioned in the methods section but is not described in the data analysis section.
Response: In the line 130-131 of the methods section was explained how to calculate and interpretate the values.
- Line 275: Based on the information in Lines 127—129, the average difficulty reading score is a perspicuity value between 55-65. Is this equivalent to a reading level of 5thgrade? I wonder if it would be better if the survey were in the easy range.
Response: According to the analyzes of Szigriszt Pazos, these values ​​correspond to a standard reading level and difficulty. Therefore, any person with a minimum sociocultural level can understand and interpret the items.
- Section 3.31: Consider putting these values in a table and summarize in text. It is challenging to read all these results.
Response: Thank you for this valuable comment. The table 2 was implemented to make the text more accessible.
- Line 336: Please consider inserting a heading for reliability
Response: The heading for reliability was implemented.
- Line 337: Please provide information on what is an acceptive value for a reliability coefficient in the data analysis section.
Response: The reliability coefficients and interpretation were explained in lines 245 to 250.
- Line 385: Shouldn’t the instrument be both reliable and valid?
Response: An instrument used in research should be both reliable and valid to ensure the accuracy of the data collected. Reliability refers to the consistency/stability of the instrument when it measures the same concept under similar conditions. Validity refers to whether the instrument measures what it is intended to measure. An instrument can be reliable (consistent) but still not be valid if it does not measure the intended concept accurately. Together, reliability and validity ensure the instrument provides consistent and accurate data, which is essential for drawing meaningful conclusions from the research.
Reviewer 2 Report
Comments and Suggestions for Authors
Dear editor
Thanks for giving me the chance to review this manuscript.
Introduction
The introduction needs to make more clarifications on the knowledge gap necessities the need of this study.
What are the previous tools discussed this issue?
From your tools:
You mean by reproductive and sexual rights, health care rights, because reproductive and sexual rights are much wider than that you mentioned in your tool. You may change the tool name to women's knowledge about healthcare rights during maternity cycle.
For resource scarcity tool, items number 3,7,8 and 9 are not related to resources. They are related to psychological and social adaptation to pregnancy.
Do you think that the abbreviation of" MatER" is suitable for perception of resource scarcity? Why it is suitable
Do you think that the abbreviation of" MatCODE" knowledge that women have of their healthcare rights during pregnancy, labor or postpartum.? Why it is suitable
Methodology:
Please, elaborate on the perception of resource scarcity (MatER)
From your inclusion criteria (The inclusion criteria were as follows: >18 139 years, having vaginal labor or C-section in the last three years, have received healthcare for the last pregnancy, labor and early postpartum (up to 40 days after labor), and good Spanish language understanding.)
The woman who is postpartum more than 40 days, still included because ahe have labor in the past three years. Why you exclude woman in the postpartum period more than 40 days?
Why it is important to put the participants ≥mean or <mean inside the participants' flowchart?
In line 172 and 173, " Firstly, it was obtained sociodemographic and obstetric variables, and validated psychometric test and, secondly the MatCODE and MatER questionnaires."
What do you mean by " validated psychometric test" as a part of your questionnaire? Line number 172
In line 208," Qualitative variables were expressed in relative frequencies (%) "
How can qualitative variables be expressed in frequency and percentage?
There are numerous typo-errors all over the article.
Author Response
Response: Thank you for your time spend reviewing our article. Your comments were carefully reviewed and were answer point-by-point below.
Introduction. The introduction needs to make more clarifications on the knowledge gap necessities the need of this study. What are the previous tools discussed this issue?
Response: New inside was written in the introduction section to clarify the importance to develop new tools in this field.
From your tools:
You mean by reproductive and sexual rights, health care rights, because reproductive and sexual rights are much wider than that you mentioned in your tool. You may change the tool name to women's knowledge about healthcare rights during maternity cycle.
Response: Thank you for this comment, with which we agree. Therefore, we have changed the name of the scale to women's knowledge of healthcare rights during maternity.
For resource scarcity tool, items number 3,7,8 and 9 are not related to resources. They are related to psychological and social adaptation to pregnancy.
Response: This comment is very relevant and need to be clarified. We thought that psycho-emotional strategies are part of the coping that woman would have with its challenges during motherhood, being an aspect of psycho-emotional resources that is not explored in other scales. The items 3, 7, 8 and 9 are related to internal resources, being the other items refer to external resources. Adaptation to changes, such as motherhood, improve based on the mood stability (DOI: 10.1002/imhj.21692), the absence of physical health problems (DOI: 10.1136/bmjopen-2021-050287; DOI: 10.15174/au.2014.724), motivation (DOI: 10.4321/s0211-573520200020003), and functionality of cognitive processes, which impacts how the woman processes the information received about motherhood (DOI: 10.1016/j.srhc.2018.06.003). For these reasons, we thought that psycho-emotional strategies are part of the coping that woman would have with its challenges during motherhood, being an aspect of psycho-emotional resources that is not explored in other scales. A clarification of the scale was applied to line 100-101.
Do you think that the abbreviation of "MatER" is suitable for perception of resource scarcity? Why it is suitable. Do you think that the abbreviation of "MatCODE" knowledge that women have of their healthcare rights during pregnancy, labor or postpartum.? Why it is suitable
Response: The authors think that the abbreviations are appropriate, since they were originally validated in the Spanish language. MaterCODE refers to "Mater-nidad" (Maternity) and “COnocimiento” (Knowledge) and “DErechos” (Rights). MatER refers to " Mat-ernidad " and “Escasez” (Scarcity) and “Recursos” (resources), something that the authors want to maintain as an imprint of the Spanish-scales.
Methodology: Please, elaborate on the perception of resource scarcity (MatER)
Response: A clarification on the perception of resource scarcity was written in the methodology (lines 96-100).
From your inclusion criteria (The inclusion criteria were as follows: >18 years, having vaginal labor or C-section in the last three years, have received healthcare for the last pregnancy, labor and early postpartum (up to 40 days after labor), and good Spanish language understanding). The woman who is postpartum more than 40 days, still included because she has labor in the past three years. Why you exclude woman in the postpartum period more than 40 days?
Response: We wanted to evaluate these scales in those women who have remembered their postpartum experience. As time passes, it can lose details of what really happened during childbirth, a key circumstance for women in the fulfillment of rights, and in the assessment of your support resources. Healthcare during the first month of postpartum is key to the perception of maternity rights (DOI: 10.1186/s12884-023-05813-0). Furthermore, this criterion is related to whether the woman has received professional healthcare during pregnancy, childbirth and postpartum. Since there are women who decide whether or not they can go to a health center. The intention is to provide tools that health professionals have prevention mechanisms.
Why it is important to put the participants ≥mean or <mean inside the participants' flowchart?
Response: According to STARD checklist that it was used as a guideline for this observational study, it needs to define the test cut-offs of the index test and account how many participants follow this criterion (https://www.equator-network.org/reporting-guidelines/stard/).
In line 172 and 173, " Firstly, it was obtained sociodemographic and obstetric variables, and validated psychometric test and, secondly the MatCODE and MatER questionnaires." What do you mean by "validated psychometric test" as a part of your questionnaire? Line number 172.
Response: Sorry for this misunderstanding. First, the participants' responses to the sociodemographic variables and the questionnaires such as PANAS or RS-14 were collected. Subsequently, the women's responses to the MatCODE and MatER were collected. The sentences were re-written.
In line 208, "Qualitative variables were expressed in relative frequencies (%)" How can qualitative variables be expressed in frequency and percentage?
Response: Qualitative variables, also known as categorical variables, can be expressed in terms of frequency and percentage by counting how often each category occurs and then calculating its proportion relative to the total. To avoid confusion, “qualitative” was modified by “categorical”.
There are numerous typo-errors all over the article.
Response: The text has been carefully reviewed to detect and correct these errors.
Round 2
Reviewer 1 Report
Comments and Suggestions for Authors
This is an interesting topic. This tool would be helpful in a research setting.
Comments on the Quality of English Language
I could not follow much of the paper. Please have someone well-versed in English edit the paper.
Author Response
This is an interesting topic. This tool would be helpful in a research setting.
Response: Thank you for the comment and your time evaluating our manuscript. The authors are planning a study to determine use in research setting for different cultural contexts.
I could not follow much of the paper. Please have someone well-versed in English edit the paper.
Response: We have reviewed the entire document again for proper understanding. If it detects errors, please let us know and we will be happy to correct them.